# A Novel Automated Chemiluminescence Method for Detecting Cerebrospinal Fluid Amyloid-Beta 1-42 and 1-40, Total Tau and Phosphorylated-Tau: Implications for Improving Diagnostic Performance in Alzheimer’s Disease

**DOI:** 10.3390/biomedicines10102667

**Published:** 2022-10-21

**Authors:** Marina Arcaro, Chiara Fenoglio, Maria Serpente, Andrea Arighi, Giorgio G. Fumagalli, Luca Sacchi, Stefano Floro, Marianna D’Anca, Federica Sorrentino, Caterina Visconte, Alberto Perego, Elio Scarpini, Daniela Galimberti

**Affiliations:** 1Neurodegenerative Diseases Unit, Fondazione IRCCS Ca’ Granda, Ospedale Maggiore Policlinico, 20122 Milan, Italy; 2Department of Pathophysiology and Transplantation, Dino Ferrari Center, University of Milan, 20122 Milan, Italy; 3Department of Biomedical, Surgical and Dental Sciences, Dino Ferrari Center, University of Milan, 20122 Milan, Italy; 4Fujirebio Italia S.r.l.-Via Pontina Km 29, 00071 Pomezia, Italy

**Keywords:** CSF, biomarkers, Alzheimer’s disease, ELISA, CLEIA

## Abstract

Recently, a fully automated instrument for the detection of the Cerebrospinal Fluid (CSF) biomarker for Alzheimer’s disease (AD) (low concentration of Amyloid-beta 42 (Aβ42), high concentration of total tau (T-tau) and Phosphorylated-tau (P-tau181)), has been implemented, namely CLEIA. We conducted a comparative analysis between ELISA and CLEIA methods in order to evaluate the analytical precision and the diagnostic performance of the novel CLEIA system on 111 CSF samples. Results confirmed a robust correlation between ELISA and CLEIA methods, with an improvement of the accuracy with the new CLEIA methodology in the detection of the single biomarkers and in their ratio values. For Aβ42 regression analysis with Passing–Bablok showed a Pearson correlation coefficient r = 0.867 (0.8120; 0.907% 95% CI *p* < 0.0001), T-tau analysis: r = 0.968 (0.954; 0.978% 95% CI *p* < 0.0001) and P-tau181: r = 0.946 (0.922; 0.962 5% 95% CI *p* < 0.0001). The overall ROC AUC comparison between ROC in ELISA and ROC in CLEIA confirmed a more accurate ROC AUC with the new automatic method: T-tau AUC ELISA = 0.94 (95% CI 0.89; 0.99 *p* < 0.0001) vs. AUC CLEIA = 0.95 (95% CI 0.89; 1.00 *p* < 0.0001), and P-tau181 AUC ELISA = 0.91 (95% CI 0.85; 0.98 *p* < 0.0001) vs. AUC CLEIA = 0.98 (95% CI 0.95; 1.00 *p* < 0.0001). The performance of the new CLEIA method in automation is comparable and, for tau and P-tau181, even better, as compared with standard ELISA. Hopefully, in the future, automation could be useful in clinical diagnosis and also in the context of clinical studies.

## 1. Introduction

Several studies report the usefulness of cerebrospinal fluid (CSF) biomarkers in the diagnostic setting of Alzheimer’s disease (AD) [1] and recent evidence underline an important association between CSF biomarkers such as Amyloid-beta 1-42 (Aβ42), tau and AD neuropathological changes (ADNC) [2].

The biomarker pattern, commonly referred to as the “AD signature”, typically displays decreased concentration of Aβ42 and increased concentration of total tau (T-tau) and Phosphorylated-tau (P-tau181). In particular, by combining CSF Aβ42, T-tau and P-tau181, a higher diagnostic accuracy for identification of AD from non-AD dementia, as well as the prediction of progression to AD in patients with Mild Cognitive Impairment (MCI), can be reached [3]. Moreover, an increased accuracy was acquired by Aβ 1-40 (Aβ40) in association to Aβ42: Aβ42/Aβ40 ratio better correlates with both amyloid plaques and imaging at Positron Emission Tomography (PET) with Aβ tracer. This ratio also reduces typical intra and inter-individual biological variability, improving the diagnostic performance, and also enabling better discrimination among different disease severity [4,5,6]. Therefore, the combination of CSF biomarkers (Aβ42, Aβ40 T-tau, P-tau181) and their ratio (Aβ42/Aβ40 or also T-tau/Aβ42, P-tau181/Aβ42) increase the performance and allow AD diagnosis at earlier stages of disease [7].

In the last few years, enzyme-linked immunosorbent assays (ELISA) were mostly used for the quantification of Aβ42, T-tau and P-tau181 [3,8,9,10]. This method, which requires manual reagent addition and removal on multi-well plates, could also be semi-automated by using ELISA-processors. However, the widespread implementation in routine clinical labs is so far difficult. In this scenario, it is important to underline that the lack of standardization, the awareness of the importance of the pre-analytics and proper specimen management, as well as the analytical factors that impact the final result of the assays.

Recently, a fully automated instrument for the detection of these biomarkers has been implemented, aiming to further improve the reproducibility and the sensitivity of the measurements. The Fujirebio Lumipulse system is a complete automatic instrument based on Chemiluminescent Enzyme Immuno Assay (CLEIA). The importance of the automation with the Lumipulse system has recently been considered from the Biofluid Based Biomarkers Professional Interest Area (BBB-PIA) working group of the Alzheimer’s Association. It was highlighted that 76.5% of participants used the Lumipulse system and overall, 88.2% of experimenters took advantage of automated platforms [11].

This technology has been already used in clinical routine in non-neurological settings and it has shown consistent inter-assay measures in serum [12,13]. Nevertheless, few studies investigate the precision of this system in the detection of CSF biomarkers and only limited studies about methods’ comparison and diagnostic performance, particularly in new innovative platforms, are available [14,15,16].

In this framework, we conducted a comparative analysis between ELISA and CLEIA methods in order to evaluate the analytical precision and the diagnostic performance of the novel CLEIA system. In particular, we measured, with both methods, Aβ42, T-tau and P-tau181 levels in a large set of CSF samples in order to compare the two analytical methods. Furthermore, we evaluated, in a fraction of samples, the clinical performance to establish an optimal threshold of discrimination of the categories with Lumipulse assay.

## 2. Materials and Methods

### 2.1. Patients

The population consisted of 111 subjects (67 males, 44 females, mean age 75 years, range 67–83) who were admitted to the Neurodegenerative Diseases Unit of the Fondazione Ca’ Granda, IRCCS Ospedale Maggiore Policlinico, University of Milan (Milan, Italy) between July 2019 and December 2020. The clinical workup included detailed past medical history, general and neurological examination, routine blood tests, formal neurocognitive assessment, CSF biomarkers Aβ42, T-tau and P-tau181 determination, brain computed tomography (CT) scan or magnetic resonance imaging (MRI), and, if needed, [18F]-fludeoxyglucose positron emission tomography. The presence of significant vascular brain damage was excluded (Hachinski Ischemic Score < 4). Lumbar punctures were performed after one night of fasting. Thirty one patients were diagnosed with AD according to the criteria of the International Working Group (IWG)-2 guidelines [17], 49 patients were diagnosed with other non-AD dementias [18,19], 31 subjects underwent LP in suspicion of a neurodegenerative disease but were discharged with no evidence of such diseases. Moreover, MMSE at time of LP was ≥28 and they did not worsen over a 12/18-month follow up, thus were considered controls.

This study was approved by the Institutional Review Board of the Fondazione Ca’ Granda, IRCCS Ospedale Maggiore Policlinico (Milan, Italy). All patients and or their caregivers gave their written informed consent.

The demographic data of patients are described in Table 1.

### 2.2. CSF Collection

CSF samples were collected into 15 mL polypropylene tubes by LP in the L3/L4 or L4/L5 interspace. The LP was conducted between 8 and 10 a.m. after one-night fasting. Following LP, CSF samples were centrifuged at 2000 r/min for 10 min at 4 °C. The supernatants were aliquoted in polypropylene tubes and stored at −80 °C until use.

### 2.3. ELISA Assay

In the morning, one aliquot of CSF was thawed at room temperature for each sample. Aβ42, T-tau and P-tau181 were measured using, respectively, three commercially available sandwich enzyme-linked immunosorbent assay (ELISA) kits, INNOTEST Amyloid-beta 42, Tau and P-Tau 181 assays (INNOTEST Fujirebio, Ghent, Belgium) according to the instructions of the manufacturer. A dedicated microplate and relative calibrators, two Run Validation Controls (RVC) as High and Low Controls, specific antibodies (two IgG specific antibodies labelled to biotin) for each analyte and relative working diluent solutions, wash, substrate and stop buffers were provided for each ELISA assay. In particular, for the calibration, 6 points levels were used in a targeted concentration range between 62.5–4000 pg/mL for Aβ42, 50–2500 pg/mL for Tau, 15.6–1000 pg/mL for P-tau181 and, together with RVC and samples, were uploaded in duplicates wells in the microplate. A specific IgG labelled with biotin in phosphate buffer called Conjugate 1 was also added. after time of incubation, a cycle of washing to remove any unbound substances. Another IgG biotin-conjugated specific antibody called Conjugate 2, that detected the first antigen-antibody complex by a Peroxidase-labeled Streptavidin, was added. Then, after other incubation and washing, a Substrate solution Tetramethyl benzidine (TMB) was uploaded. The color developed was in proportion to the amount of protein bound in the initial step: the reaction was stopped with Stop Solution and the intensity of the color was measured using a microplate reader set to a wavelength of 450 nm so protein’s levels were calculated by an interpolated standard curve. The Limit of Detection (LoD) was 65 pg/mL for Aβ42, 34 pg/mL for T-tau and 13 pg/mL for P-tau181 according to the manufacturer’s instructions.

### 2.4. CLEIA Assay

The new automated chemiluminescence enzyme immunoassay (CLEIA) method (Lumipulse G600II System, Fujirebio, Tokyo, Japan) was used for the measurement of Aβ42, Aβ40, T-tau and P-tau181 in the CSF samples, using respective Lumipulse assays (Lumipulse Aβ42, Lumipulse Aβ40, Lumipulse T-tau, Lumipulse P-tau181 Immunoreaction Cartridges with the same ELISA’s antibodies and the same Substrate, Diluent and Wash Solutions reagents, Fujirebio, Ghent, Belgium). All measures were performed in the same batch of samples in run and single of reagents, calibration and three different internal controls (High, Medium and Low Levels) were processed at the beginning to ensure that all measured values testing the correct immunoassay functionality and were in the right range.

For the calibration, a three-point master curve was used with a range respectively between 0 and 3409 pg/mL for Aβ42 and between 0 and 30,000 pg/mL for Aβ40, a range between 0 and 2250 pg/mL for T-tau and between 0 e 400 pg/mL for P-tau181. All reagents and samples were uploaded on the roundabout of the automatic platform and results of the assay were provided on touchscreen in real-time at the end of each sample’s assays, after transferable on printer and/or USB system. The LoD was 7.17 pg/mL for Aβ42, 2.78 pg/mL for Aβ40, 141 pg/mL for T-tau, 0.282 pg/mL for P-Tau181 according to the manufacturer.

### 2.5. Statistical Analysis

All statistical analysis was performed by Excel Analyse-it ^®^ v. 5.90 (Leeds, UK). Non-parametric Passing-Bablok regression was used for ELISA and CLEIA methods comparison; it is a statistical procedure that allows a reliable estimation of analytical methods agreement and also of possible systematic bias between them. This method is robust, non parametric and non sensitive to distribution of errors and data outlier [20]. Regression analysis allows you to evaluate the error constant systematic (intercept) and proportional (slope). Assumption for proper application of Passing–Bablok regression involves variables continuously distributed and linear relationship among data measured by two analytical methods. The results are shown by scatter diagram and regression line, and regression equation where intercept represents constant and slope proportional measurement error. The existence of systematic and proportional differences between the two methods was assessed through the 95% confidence intervals (CI) of the intercepts and slopes of the regression equations. The slope and intercept explain also if values differ only by chance, allowing conclusion of the method agreement.

Diagnostic performance of Lumipulse assays for the CSF biomarker’s ratio used to distinguish between AD patients and controls was assessed by means of a receiver operating characteristic (ROC) curve analysis. Optimal thresholds were determined by maximizing the Youden index and sensitivity, and specificity was calculated. The ROC curves were compared according to the area under the curve (AUC) comparison method of DeLong et al. [21].

## 3. Results

We performed ELISA and CLEIA on the same samples and on the same day, in order to avoid environmental bias. CSF Aβ42, T-tau and P-tau181 measurements by using both methods were completed successfully for all 111 subjects.

### 3.1. Passing-Bablok Regression Analysis in Method Comparison between INNOTEST and Lumipulse Assays

Mean ± SEM levels of the three analytes detected with ELISA as compared with CLEIA were as follows: Aβ42 776 ± 32 vs. 763 ± 26 pg/mL (*p* > 0.05), T-tau: 521 ± 40 vs. 552 ± 37 pg/mL (*p* > 0.05), P-tau181: 66 ± 3 vs. 88 ± 6 pg/mL (*p* > 0.05). No significant differences between concentrations obtained with the two methods for Aβ42, T-tau and P-tau181 were shown.

Of note, Aβ42 regression analysis in our cohort showed a Passing–Bablok regression with a Pearson correlation coefficient r = 0.867 (*p* < 0.0001, 95% CI: 0.8120–0.907) intercept =−59.47, slope 0.80 (Figure 1A). Regarding T-tau analysis, we observed a Pearson correlation coefficient r = 0.968 (*p* < 0.0001, 95% CI 0.954–0.978) intercept = 55.62, slope = 0.97 (Figure 1B). For P-tau181, we observed a Pearson correlation coefficient r = 0.946 (*p* < 0.0001, 95% CI 0.922–0.962), intercept = −33.56, slope = 1.81 (Figure 1C).

### 3.2. Diagnostic Performance of Lumipulse Using ROC Analysis between Both Methods

To investigate the diagnostic performance of the new automated method, ROC analysis was assessed. To this aim, a training clinical cohort of 31 AD and 31 controls was considered, in order to determine an optimal threshold for the new method. The overall ROC AUC comparison between ROC in ELISA and ROC in CLEIA confirmed a more accurate ROC AUC with the new automatic method: T-tau AUC ELISA = 0.94 (*p* < 0.0001, 95% CI 0.89–0.99) vs. AUC CLEIA = 0.95 (*p* < 0.0001, 95% CI 0.89–1.00), and P-tau181 AUC ELISA = 0.91 (95% CI 0.85; 0.98 *p* < 0.0001) vs. AUC CLEIA= 0.98 (*p* < 0.0001, 95% CI 0.95; 1.00). On the contrary, for Aβ42, it was found an ELISA ROC Aβ42 AUC = 0.98 (*p* < 0.0001, 95% CI 0.95–1.00) better than CLEIA Aβ42 ROC AUC = 0.92 (*p* < 0.0001, 95% CI 0.86–0.99) but a relevant improvement was observed by the introduction of the Aβ42/Aβ40 ratio (CLEIA Aβ42/Aβ40 ROC AUC = 0.98, *p* < 0.0001, 95% CI 0.96–1.00) (Figure 2A,B).

In an effort to reach more accuracy, CLEIA ratios were considered also for T-tau/Abeta42 and P-tau181/Aβ42. Considering the former, ROC AUC was 0.98, whereas for the latter ROC AUC was 0.99, showing an improvement of the performance (Figure 2C). Then, basing on ROC analyses showed above and subsequent Youden Index, we calculated the new optimal thresholds, which were 544 pg/mL for Aβ (sensitivity 94%, specificity 81%) and 0.055 for Aβ42/Aβ40 (sensitivity 90%, specificity 90%). The optimal threshold for T-tau was 402 pg/mL (sensitivity 87% specificity 94%), whereas for P-tau181 was 60.5 pg/mL (sensitivity 90%, specificity 90%).

Optimal thresholds were calculated for T-tau/Aβ42 and P-tau181/Aβ42 ratios: 0.75 (sensitivity 97%, specificity 97%) and 0.13 (sensitivity 97%, specificity 94%), respectively, further increasing the accuracy. Table 2 summarizes optimal thresholds, together with related 95% confidence intervals, obtained for the various analytes with the two different tests.

## 4. Discussion

Herein, we demonstrated that there is a robust correlation between the ELISA and CLEIA methods, with an improvement of the accuracy with the new CLEIA methodology in the detection of the single biomarkers and in their ratio values. Significant correlations between the two different methodological procedures were demonstrated for Aβ42, T-tau and P-tau181 analyses. Regarding Aβ42 levels, we confirmed that the linear relationship is stronger for lower levels compared to higher ones, as previously observed [22]. Conversely, regarding T-tau and P-tau181 levels, we established a good correlation in the values spanning the whole range, whereas for P-tau181, it remained stronger for lower levels.

We also showed that ROC analysis suggested optimal thresholds applicable for almost all the biomarkers and for their ratio. In line with the existing literature [14,22], with the new method, introducing Aβ42/Aβ40 lead to an improvement of the accuracy. The same result was obtained with the introduction of the ratios for T-tau/Aβ42, P-tau181/Aβ42. Unfortunately, we could not perform the Aβ42/Aβ40 ratio in the retrospective cohort because the Aβ40 dosage in ELISA was not carried out.

Nowadays, CSF biomarkers tests are for use in routine clinical practice, CSF biomarker profile supports the diagnosis of AD in terms of amyloid and tau biomarkers; CSF ratio tests provide to improve the accuracy also better of an imaging test for early, rapid, easy detection and cheaper costs. Recent findings report different cases in which CSF biomarkers are useful at different stages of AD diagnosis and in different ages; in particular, CSF Aβ42 assays have good agreement with amyloid PET imaging, while Aβ42/Aβ40 and tau/Aβ42 ratios have superior performance to Aβ42 alone [23,24].

Here, we did not consider clustering approach [25] allowing to include all samples (i.e., all demented patients) in the analysis because the cohort is quite small, particularly also the group of other dementias, to foresee relevant results.

Our results demonstrate the potential value of an automated method for CSF biomarker determination, that will reduce analytical errors from the operator intervention, thus increasing the diagnostic performance. Besides reference values for each biomarker, the Aβ42/Aβ40 together with other ratios increase the accuracy and would be of help for a matter of reproducibility among different centers [26,27]. In fact, AUC ROC determination and sensitivity and specificity improve with the Aβ40 introduction or in T-tau/Aβ42, P-tau181/Aβ42 calculation. However, an analysis in a larger validation cohort, preferably on patients with AD neuropathological demonstration, would be needed to confirm our data in particular to validate the thresholds identified.

Relevance of this advanced method remains above all the high-throughput, contemporary and full in automation analyzing of the biomarkers. Automation offers a higher level of standardization, combining multiple parameters in comparison to manual testing. Therefore, the novel CLEIA method allows to analyze, in automation and quickly, different biomarkers in each sample, therefore it reduces the variability due to the pre-analytical factors: it is also of strong relevance as a faster and easy management in clinical settings, with a reduction of costs and high-resolution method analysis. Lumipulse demonstrated indeed an outstanding performance in terms of precision, linearity, analytical and functional sensibility, intra-inter assay variability, so these peculiar hallmarks emphasize its better involvement to test reproducibility [28].

CSF biomarkers determination has shown great utility in the prediction of AD pathology, however the standardization and simplification of biomarker determination methods still remains a fundamental issue in clinical practice. Despite with Lumipulse many limits are overcome, there are still some features to better refine.

Pre-analytical and analytical variability of CSF biomarkers for the diagnosis of AD and related dementia hinder their diffusion in routine and clinical settings, as well as the definition of universally recognized threshold values. Therefore, it is necessary to examine laboratory procedures that potentially contribute to this variation together with the validation of new diagnostic values of biomarkers in large multi-center studies. CSF collection and handling potentially contribute to the existing variation in addition to validating the diagnostic value of biomarkers [29].

Moreover, recent literature underlines an incipient role of peripheral biomarkers (in particular blood based biomarkers as plasma P-tau epitopes, P-tau181, P-tau217, P-tau231, NfL and GFAP [30] but also other peripheral noninvasive biomarkers as plasma micro RNAs (miRNAs) profiling also in terms of Extracellular Vesicles’ “cargo” [31] that track different aspects of the disease as synaptic dysfunction, neuro-inflammation, and glial activation, useful for early diagnosis of MCI, AD and differential diagnosis of AD from other neurodegenerative diseases. In particular, a new promising scenario of innovative platform panels could bring to set new diagnostical parameters and typical thresholds for the analytes and digital tools may also contribute better to screening and diagnostic pathways in AD according to a novel AT(N) definition in Alzheimer continuum [32,33,34].

## 5. Conclusions

Herein, we showed that the performance of the new CLEIA method in automation is comparable and, for T-tau and P-tau181, even better, as compared with standard ELISA in our population. Thanks to Lumipulse, as only few data reported today, a significant increase in the linearity and reproducibility in the measures at first was assessed successfully as well as a better accuracy through ratio determination. Hopefully, in the future, automation could be useful in clinical diagnosis and also in the setting of clinical studies especially also with the introduction of the new promising peripheral blood-based biomarkers as prescreening tools in clinical trials and in large scale population of epidemiological studies.

## Figures and Tables

**Figure 1 biomedicines-10-02667-f001:**
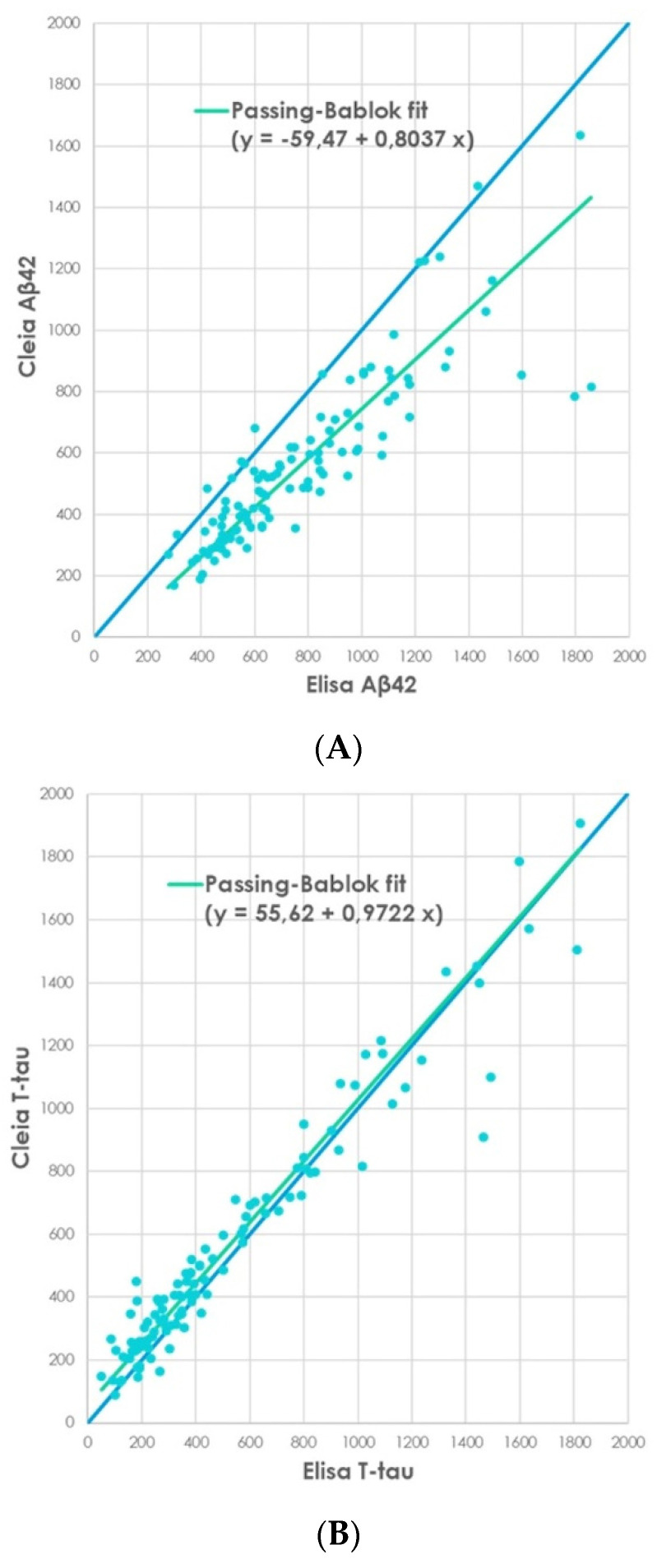
Correlation analysis by non-parametric Passing–Bablok regression for method comparison between classical manual ELISA and Lumipulse assay for all 111 subjects: (**A**) Aβ42, (**B**) T-tau, (**C**) P-tau181.

**Figure 2 biomedicines-10-02667-f002:**
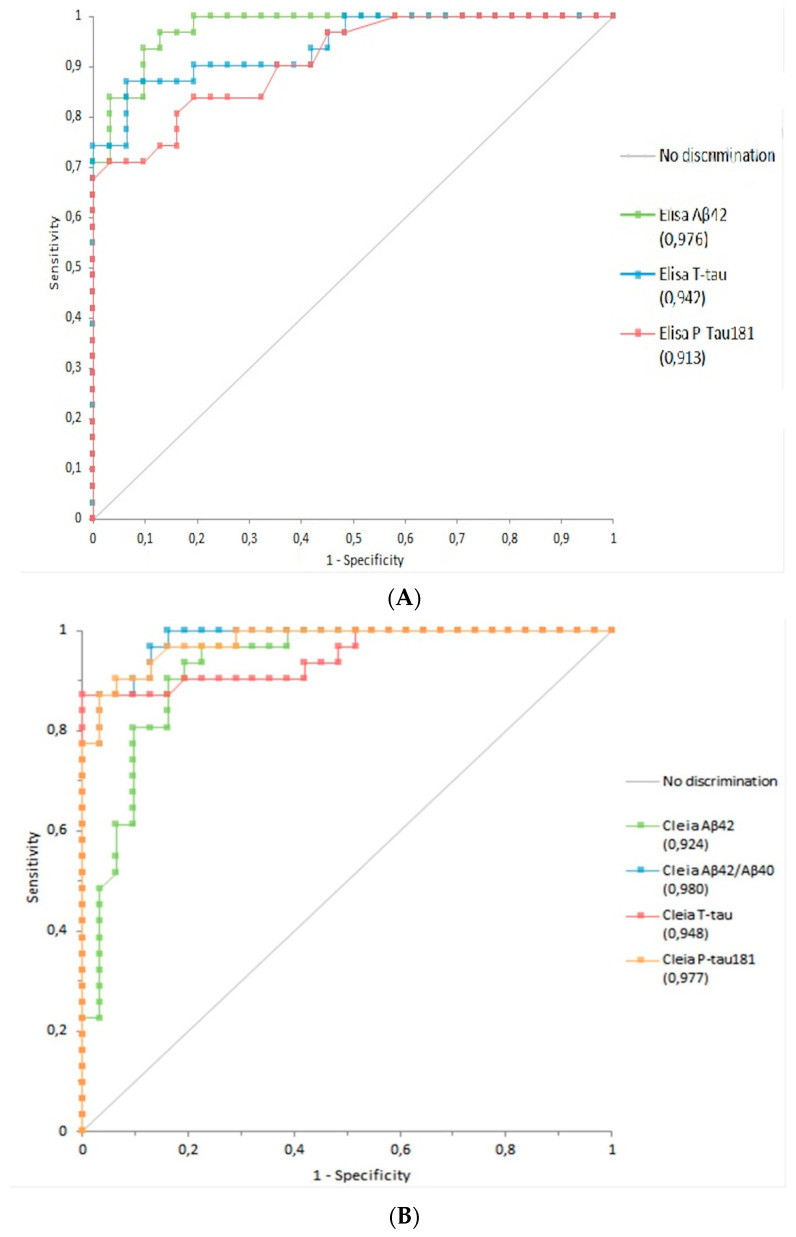
Receiver operating characteristic (ROC) curves to distinguish between AD patients and controls: (**A**) AUC analysis for ELISA Aβ42, T-tau and P-tau181 (**B**) AUC analysis for CLEIA Aβ42, Aβ42/Aβ40, T-tau and P-tau181, (**C**) AUC analysis for ratio T-tau/Aβ42 and P-tau181/Aβ42.

**Table 1 biomedicines-10-02667-t001:** Characteristics of the population.

	AD	Controls	Other Dementia
N	31	31	49
Gender (F/M)	14/17	10/21	15/34
Age (years ± SEM)	75 ± 2	73 ± 1	75 ± 1
CSF Aβ42 ELISA (pg/mL ± SEM)	520 ± 15	1063 ± 65	756 ± 41
CSF Aβ42 CLEIA (pg/mL ± SEM)	379 ± 21	788 ± 55	683 ± 50
CSF T-tau ELISA (pg/mL ± SEM)	603 ± 77	233 ± 16	651 ± 74
CSF T-tau CLEIA (pg/mL ± SEM)	610 ± 37	285 ± 16	683 ± 71
CSF P-tau181 ELISA (pg/mL ± SEM)	72 ± 4	43 ± 3	77 ± 6
CSF P-tau181 CLEIA (pg/mL ± SEM)	105 ± 6	42 ± 2	105 ± 12
CSF Aβ42/Aβ40 CLEIA	0.043 ± 0.002	0.081 ± 0.003	0.060 ± 0.003
CSF T-tau/Aβ42 ELISA	1.189 ± 0.001	0.243 ± 0.001	0.076 ± 0.008
CSF T-tau/Aβ42 CLEIA	1.727 ± 0.133	0.402 ± 0.036	1.806 ± 0.290
CSF P-tau181/Aβ42 ELISA	0.142 ± 0.009	0.045 ± 0.004	0.147 ± 0.023
CSF P-tau181/Aβ42 CLEIA	0.295 ± 0.021	0.061 ± 0.007	0.011 ± 0.001

**Table 2 biomedicines-10-02667-t002:** CSF optimal thresholds and related 95% confidence intervals for ELISA and CLEIA.

Biomarkers	AUC 95% CI AD vs. Controls	Sensitivity 95% CI AD vs. Controls	Specificity 95% CI AD vs. Controls	Thresholds 95% CI AD vs. Controls
ELISA Aβ42	0.98 (0.95, 0.10)	0.90 (0.78, 0.99)	0.84 (0.64, 0.91)	**533** (521, 550) pg/mL
CLEIA Aβ42	0.92 (0.86, 0.99)	0.94 (0.79, 0.99)	0.81 (0.64, 0.91)	**544** (543, 550) pg/mL
CLEIA Aβ42/Aβ40	0.98 (0.96, 0.10)	0.90 (0.75, 0.97)	0.90 (0.75, 0.97)	**0.055** (0.053, 0.058)
ELISA T-tau	0.94 (0.89, 0.99)	0.81 (0.57, 0.86)	0.94 (0.79, 0.99)	**365** (353, 373) pg/mL
CLEIA T-tau	0.95 (0.89, 0.10)	0.87 (0.72, 0.95)	0.94 (0.79, 0.99)	**402** (398, 404) pg/mL
ELISA P-tau181	0.91 (0.85, 0.98)	0.81 (0.57, 0.86)	0.94 (0.67, 0.95)	**55** (54.5, 57.0) pg/mL
CLEIA P-tau181	0.98 (0.95, 0.10)	0.90 (0.75, 0.97)	0.90 (0.72, 0.95)	**60.5** (58.5, 63.0) pg/mL
ELISA T-tau/Aβ42	0.99 (0.97, 0.10)	0.97 (0.83, 0.10)	0.94 (0.79, 0.99)	**0.47** (0.42, 0.84)
CLEIA T-tau/Aβ42	0.98 (0.97, 0.10)	0.97 (0.84, 0.10)	0.97 (0.79, 0.99)	**0.75** (0.72, 0.84)
ELISA P-tau181/Aβ42	0.98 (0.97, 0.10)	0.96 (0.89, 0.10)	0.90 (0.79, 0.99)	**0.07** (0.04, 0.08)
CLEIA P-tau181/Aβ42	0.99 (0.97, 0.10)	0.97 (0.83, 0.10)	0.94 (0.79, 0.99)	**0.13** (0.11, 0.16)

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
