# Peer review of "A Novel Automated Chemiluminescence Method for Detecting Cerebrospinal Fluid Amyloid-Beta 1-42 and 1-40, Total Tau and Phosphorylated-Tau: Implications for Improving Diagnostic Performance in Alzheimer’s Disease"

_biomedicines, 2022, doi:10.3390/biomedicines10102667_

Round 1

Reviewer 1 Report

The manuscript proposed by Arcaro, Fenoglio and co-workers deals with the demostration of equivalence of ELISA ans CLEIA system for detecting Abeta42, P-tau and P-tau181 in a large setting of CSF samples. The adavantages of CLEILA system, especially the possibility to avoid the variability due to the pre-analytical factors, make it a trustable diagnostic method for AD. 

Author Response

"we thank the reviewer for his/her positive feedback".

Reviewer 2 Report

In their work, Arcaro et al. provided further evidence for the robustness of automated CLEIA measurements in the context of CSF biomarkers of AD. Specifically, the authors tested for concordance between CLEIA and ELISA in a cohort consisting of clinically diagnosed AD patients, controls, and patients with other neurodegenerative diseases. The work is not particularly innovative, as at least three other papers (all of which have already been correctly cited) have provided similar results. However, I find the work to be well-conducted and the results may be of interest in clinical practice. I therefore suggest some changes that could make the manuscript more informative and appealing.

1) In the introduction, it would be appropriate to provide statistics on the use of the platform mentioned in the various expert centers. In particular. I suggest citing this article that says 76.5% of participants use the platform used by the authors and in general 88% of experimenters use automated platforms (https://doi.org/10.1002/alz.12545).

2) Add PB regression for Ab42/Ab40 ratio. The ratio may be better correlated than Ab42 alone.

3) The authors performed the ROC analysis correctly by including only AD and controls. However, it should be considered, or at least recognized, that clustering approaches (such as the one described here https://www.frontiersin.org/articles/10.3389/fnins.2021.647783/fullI) may allow all samples to be included in the analysis (including other dementias). In the section on ROC analysis, it would also be useful to compare more clearly the optimal thresholds obtained for the various analytes with the two different tests. I think a table summarizing the thresholds with 95% confidence intervals (the R pROC package can be used for this, it makes use of the bootstrap method to calculate confidence intervals) would help.

Author Response

Point 1: In their work, Arcaro et al. provided further evidence for the robustness of automated CLEIA measurements in the context of CSF biomarkers of AD. Specifically, the authors tested for concordance between CLEIA and ELISA in a cohort consisting of clinically diagnosed AD patients, controls, and patients with other neurodegenerative diseases. The work is not particularly innovative, as at least three other papers (all of which have already been correctly cited) have provided similar results. However, I find the work to be well-conducted and the results may be of interest in clinical practice. I therefore suggest some changes that could make the manuscript more informative and appealing.

1) In the introduction, it would be appropriate to provide statistics on the use of the platform mentioned in the various expert centers. In particular. I suggest citing this article that says 76.5% of participants use the platform used by the authors and in general 88% of experimenters use automated platforms (https://doi.org/10.1002/alz.12545).

Response 1: thank you for this helpful suggestion. A few lines have been added to underline this point: “The importance of the automation with the Lumipulse system has recently been considered from the Biofluid Based Biomarkers Professional Interest Area (BBB-PIA) working group of the Alzheimer’s Association. It was highlighted that 76.5% of participants used the Lumipulse system and overall the 88,2% of experimenters took advantage of automated platforms [11]”. (lines 64-68 in the introduction)

Point 2: Add PB regression for Ab42/Ab40 ratio. The ratio may be better correlated than Ab42 alone

Response 2: although we definitely agree with this suggestion, unfortunately we could not perform the ratio in the retrospective cohort because the Ab40 dosage in ELISA was not carried out. We acknowledged this limitation in the discussion (lines 266-267)

Point 3: The authors performed the ROC analysis correctly by including only AD and controls. However, it should be considered, or at least recognized, that clustering approaches (such as the one described here https://www.frontiersin.org/articles/10.3389/fnins.2021.647783/fullI) may allow all samples to be included in the analysis (including other dementias). In the section on ROC analysis, it would also be useful to compare more clearly the optimal thresholds obtained for the various analytes with the two different tests. I think a table summarizing the thresholds with 95% confidence intervals (the R pROC package can be used for this, it makes use of the bootstrap method to calculate confidence intervals) would help.

Response 3: Thank you for raising this issue; a few lines have been added to underline this point:  “Here, we did not consider clustering approach [25] allowing to include all samples (i.e all demented patients) in the analysis because the cohort is quite small, particularly also the group of other dementias, to foresee relevant results”. (lines 275-277 in the discussion).

We included a table (Table 2) summarizing optimal thresholds, together with related 95% confidence intervals, obtained for the various analytes with the two different tests , together with ELISA ROC curves – up to you (Figure 2A)
